# Exact Shapley Value for Local and Global Explanation of Additive Gaussian Processes

## Abstract

Additive Gaussian Processes (AGPs) have emerged as an extension of Gaussian Processes (GPs), offering a more interpretable and flexible approach by decomposing the target function into sums of multiple GPs, each influenced by different subsets of features. Despite their enhanced, expressive structure, AGPs struggle to provide local explanations and offer only global feature importance with notable shortcomings. To bridge this gap, this paper introduces an interpretative framework for AGPs that utilizes Shapley values to provide both local and global explanations of feature importance. For local explanation, we use the relationship between the AGP and the Shapley value and guarantee the additivity of the explanation. We then develop a dynamic programming algorithm for efficient computation of *exact* Shapley values, whose complexity scales polynomially rather than exponentially with the number of features. In addition, we use a variance-based sensitivity approach for the global explanation and develop an efficient dynamic programming-based algorithm to compute the *exact* Shapley value as the global feature importance. We present the effectiveness of the proposed methods on several real experiments and discuss their potential in interpretable machine learning, feature selection, and global sensitivity analysis.

## 1 Introduction

Gaussian processes (GPs) are powerful nonparametric models widely used in machine learning for tasks like regression and classification (Rasmussen & Williams, 2006). They excel at capturing complex relationships between input features and the output, but their interpretability can be challenging due to their reliance on a single, global kernel function. Additive Gaussian Processes (AGPs) extend the GP framework by modeling the target function as a sum of multiple GPs, each representing the effect of a subset of features (Duvenaud et al., 2011). This additive structure enhances the model's ability to capture complex, higher-order interactions among features and is more effective over standard GPs in terms of flexibility and interpretability. Further, an orthogonalization technique into AGP is introduced to ensure independence among additive components, thus improving further interpretability (Lu et al., 2022). The orthogonalization leverages the functional ANOVA decomposition (Hooker, 2004), which separates a function into additive components representing the influence of any subsets of features, thus enhancing the clarity of feature importance.

From an interpretability perspective, AGPs provide substantial advantages. Duvenaud et al. (2011) used the scaling factor of interaction orders to quantify the importance of interaction order features, providing an understanding of how different features interact globally within the model. This is, however, shown to be unidentifiable (Lu et al., 2022), and a first-order Sobol index (Sobol, 2001) is calculated for the global feature importance, which measures the contribution of each feature to the overall model variance (Lu et al., 2022). Nonetheless, the first-order Sobol index fails to sum up the total variance and tends to underestimate the total contribution of a feature to the variance when there are interactions between features (as in AGP) (Owen, 2014; Song et al., 2016). Besides, the interpretation is still limited to a global context and does not provide insights into the predictions of individual instances.

To address these shortcomings, this paper explores an approach to interpretability in AGPs using Shapley values (Shapley, 1953). In particular, the contributions of this paper are two-fold. First, we propose a framework that leverages the connection between AGPs and the Shapley value, and

guarantees additive feature attribution. We then put forward an efficient dynamic programming algorithm to compute the *exact* Shapley values for local explanations, providing insight into how individual data points predictions are affected by each feature. Second, we use the variance-based global sensitivity analysis concepts and formulate the global feature importance by the Shapley value. Accordingly, we put forward another efficient dynamic programming algorithm to compute the *exact* Shapley value for the global explanation. The proposed algorithms address the shortcomings in quantifying the interpretation in AGPs and has applications in feature selection, explanation, and global sensitivity analysis.

## 2 BACKGROUND

**Notation.** In this paper, we denote the set of $d$ features as $D$, and represent its power set, which encompasses all subsets of $D$, by $2^D$. The training set containing $n$ samples is denoted by $\{\boldsymbol{x}_i, y_i\}_{i=1}^n$, where $\boldsymbol{x}_i \in R^d$ and $y \in R$. The set of all data points $\boldsymbol{x}_i$'s is denoted by $X \in R^{n \times d}$, $X_S$ refers to the training set with only features in $S$, $\mathcal{X}_S$ refers to the sample space of feature subset $S, \forall S \in 2^D$, and $p(x_i)$ refers to the density of feature $x_i$. We use capital letters for sets (except for $X$), the vectors are denoted by bold-faced lower-cased letters like $\boldsymbol{x}$, and $\boldsymbol{x}_S$ refers to only the features of $\boldsymbol{x}$ that is in $S$. We also show the element-wise product by $\odot$, and the mathematical expectation operator by $\mathbb{E}$.

### 2.1 ADDITIVE GAUSSIAN PROCESSES (AGPS)

We focus on modeling the output $y$ as a function of $d$-dimensional input features $\boldsymbol{x}$ using a hidden function $f(\boldsymbol{x})$. Duvenaud et al. (2011) introduced a GP model with an additive structure defined as:

$$f(\mathbf{x}) = \sum_{S \subseteq D} f_S(\boldsymbol{x}), \tag{1}$$

where $f_S(\boldsymbol{x})$ is a function over only the feature subset $S$. In this framework, the additive decomposition of the function is achieved by structuring the kernel accordingly. In particular, each dimension $i$ is assigned a one-dimensional base kernel $k_i(\boldsymbol{x}, \boldsymbol{x}')$ (which only operates on the $i$-th element of $\boldsymbol{x}$ and $\boldsymbol{x}'$), and the $q$-th order additive kernels are then constructed as follows (Duvenaud et al., 2011):

$$k_{add_q}(\boldsymbol{x}, \boldsymbol{x}') = \sigma_q^2 \sum_{1 \le i_1 \le i_2 \le \cdots \le i_d \le d} \left[ \bigodot_{l=1}^q k_{i_l}(\boldsymbol{x}, \boldsymbol{x}') \right], \tag{2}$$

where $\sigma_q^2$ is the variance assigned to all interactions of order $q$. The overall kernel $K$ used in AGP is constructed by summing these additive kernels up to the dimensionality of the data, i.e., $k^{add}(\boldsymbol{x}, \boldsymbol{x}') = \sum_{q=0}^d k^{add_q}(\boldsymbol{x}, \boldsymbol{x}')$ with $k^{add_0} = \sigma_0^2$. Despite exponentially many terms, a recursive method is adapted for efficient polynomial-time computation in the AGP (Duvenaud et al., 2011). Finally, the prediction of an instance like $\boldsymbol{x}$ is computed as (Rasmussen & Williams, 2006):

$$f(\boldsymbol{x}) = k^{add}(\boldsymbol{x}, X)^\top \boldsymbol{\alpha}, \qquad \boldsymbol{\alpha} = \left( k^{add}(X, X) + \sigma_n^2 I \right)^{-1} \boldsymbol{y}, \tag{3}$$

where $k^{add}(X, X)$ and $k^{add}(\boldsymbol{x}, X)$ are the additive kernel matrix and additive kernel vector over the training data $X$ and between sample $\boldsymbol{x}$ and all the training data $X$, respectively, and $\sigma_n^2$ is the noise variance in GPs.

### 2.2 ORTHOGONAL ADDITIVE GAUSSIAN PROCESSES

One challenge with AGPs is identifiability, as many basis functions can sum to $f(\boldsymbol{x})$. Durrande et al. (2011) addressed this using functional ANOVA decomposition (Hooker, 2004), imposing the following constraints on basis functions:

1. **Zero Mean:** Each component function $f_S(\boldsymbol{x})$ must have a mean of zero when averaged over all features outside its subset. Specifically, this means $\mathbb{E}[f_S(\boldsymbol{x})] = 0$ for every non-empty subset $S$, with the expectation taken over the features not included in $S$.

2. **Orthogonality:** The component functions must be orthogonal to each other. In other words, for any two distinct subsets $S \neq S'$, the condition $\mathbb{E}[f_S(\boldsymbol{x})f_{S'}(\boldsymbol{x})] = 0$ must hold, where the expectation is over the joint distribution of $\boldsymbol{x}_S$ and $\boldsymbol{x}_{S'}$.

Durrande et al. (2011) put forward a class of kernel functions that satisfy the above conditions. In particular, for a base kernel $k_i$, they define *constrained kernel* $\tilde{k}_i$ as:

$$\tilde{k}_i(\boldsymbol{x}, \boldsymbol{x}') = k_i(\boldsymbol{x}, \boldsymbol{x}') - \frac{\int k_i(\boldsymbol{x}, s)\, dp(s) \int k_i(\boldsymbol{x}, s)\, dp(s)}{\int \int k_i(s, t)\, dp(s)\, dp(t)}. \tag{4}$$

The higher-order kernels $\tilde{k}^{add_q}$ are then constructed by multiplying corresponding $\tilde{k}_i$, and the additive constrained kernel is defined as $\tilde{k}^{add}(\boldsymbol{x}, \boldsymbol{x}') = \sum_{q=0}^{d} \tilde{k}^{add_q}(\boldsymbol{x}, \boldsymbol{x}')$. A function drawn from a GP with the constrained kernel $\tilde{k}^{add}$ is then shown to satisfy the ANOVA decomposition conditions (Durrande et al., 2011). Lu et al. (2022) extend AGP with the constraint kernels and showed that for several kernels (e.g., squared exponential or categorical kernels) with a Gaussian density of features, $\tilde{k}_i$ has an analytical solution; for all other density measures or kernels, the integration in equation (4) could be estimated by empirical probability measure based on the training samples.

## 2.3 SHAPLEY ADDITIVE EXPLANATION

A class of explaining black-box predictive models is based on the Shapley value. Consider a supervised learning model $h$ trained on $\{\boldsymbol{x}_i, y_i\}$. Computing the Shapley value requires a value function $v : 2^D \to \mathbb{R}$ to quantify the payoff associated with subsets $S \subseteq D$ for a particular sample. A common choice for defining payoffs for local explanation is (Lundberg & Lee, 2017):

$$v(S) = \mathbb{E}[h(\boldsymbol{x}) \mid X_S = \boldsymbol{x}_S]. \tag{5}$$

Given a value function, the marginal contribution of $j$ to coalition $S$ at $\boldsymbol{x}$ is:

$$\Delta_v(S, j) = v(S \cup \{j\}) - v(S). \tag{6}$$

The Shapley value of feature $j$, $\phi_j$, is the weighted mean of marginal contributions over all subsets:

$$\phi_j = \sum_{S \subseteq D \setminus \{j\}} \frac{|S|!\, (d - |S| - 1)!}{d!} \left[ \Delta_v(S, j) \right]. \tag{7}$$

This formulation uniquely satisfies properties like efficiency, symmetry, sensitivity, and linearity (Shapley, 1953; Lundberg & Lee, 2017). Such an explanation is proven to decompose the model predictions into the Shapley values of features (Lundberg & Lee, 2017):

$$h(\boldsymbol{x}) = \phi_0 + \sum_{j=1}^{d} \phi_j \tag{8}$$

where $\phi_0$ is the baseline expectation $\mathbb{E}[f(\boldsymbol{x})]$ (also refered to as the value of null game $v(\{\emptyset\})$) and $\phi_j$ is the Shapley value for feature $j$. Since computing the exact Shapley value requires realizing $v$ for exponentially many subset $S \subseteq D$, it is typically approximated from a number of such subsets by a regression model (Lundberg & Lee, 2017) or a Monte Carlo method (Štrumbelj & Kononenko, 2014; Song et al., 2016) in practice.

## 3 EXACT SHAPLEY VALUE FOR LOCAL EXPLANATION

**AGP and Shapley Value.** We now present the local explanation using the Shapley value for AGPs. Instead of defining a payoff function as in equation (5), we show a relationship between the AGPs and the Shapley value and obtain the Shapley values of features directly from a trained AGP model. The following lemma states how to estimate the Shapley value of features for an instance $\boldsymbol{x}$ from the AGP model with the constrained kernel. All the proofs are provided in Appendix A.

**Lemma 1** *Let $f(\boldsymbol{x})$ be characterized by an AGP with constrained kernel basis $\tilde{k}_i$. Then, the estimated local Shapley value for feature $i$, shown by $\hat{\phi}_i^l$, is:*

$$\hat{\phi}_i^l = \left( \sum_{S \subseteq D, i \in S} \frac{\sigma_{|S|}^2}{|S|} \bigodot_{j \in S} \tilde{k}_j(\boldsymbol{x}, X) \right)^{\top} \boldsymbol{\alpha} \tag{9}$$

We now show that estimating the Shapley values as in Lemma 1 would provide us with an additive feature attribution as in equation (8).

**Lemma 2** *Defining the value of null game $v(\{\emptyset\}) = \sigma_0 \mathbf{1}^\top \boldsymbol{\alpha}$, the explanation provided by the Shapley value as in equation* (9) *is an additive feature attribution such that*

$$f(\boldsymbol{x}) - v(\{\emptyset\}) = \sum_{i=1}^{d} \hat{\phi}_i^l. \tag{10}$$

**Recursive Formulation and Dynamic Programming.** The challenge of computing the Shapley value using equation (9) lies in the computational complexity, as the sum in equation (9) grows exponentially with the number of features. We propose a dynamic programming algorithm to compute the Shapley value more efficiently. Specifically, equation (9) can be expressed recursively as:

$$\hat{\phi}_i^l = \left( \tilde{\boldsymbol{k}}_i \bigodot \tilde{l}_i(d, 1) \right)^\top \boldsymbol{\alpha}. \tag{11}$$

Here, $\tilde{\boldsymbol{k}}_i = \tilde{k}_i(\boldsymbol{x}, X)$, $\tilde{l}_i(q, t) = \tilde{\boldsymbol{k}}_q \odot \tilde{l}_i(q-1, t+1) + \tilde{l}_i(q-1, t)$, subject to the boundary conditions $\tilde{l}_i(i, t) = \tilde{l}_i(i-1, t)$ and $\tilde{l}_i(0, t) = \frac{\sigma_t^2}{t}$. In this recursion, we first factor out $\tilde{\boldsymbol{k}}_i$, as it appears in all terms of the summation. The term $\tilde{l}_i$ represents the remaining terms in the sum that exclude $\tilde{\boldsymbol{k}}_i$. The parameters of $\tilde{l}_i$ are $q$, which selects the dimension of the features, and $t$, which replicates the weights $\sigma_t^2 / t$ in the summation.

The recursion divides the terms into two disjoint sets: those that include $\tilde{\boldsymbol{k}}_q$ and those that do not. The term $\tilde{\boldsymbol{k}}_q \odot \tilde{l}_i(q-1, t+1)$ captures the terms where $\tilde{\boldsymbol{k}}_q$ is present, and we recursively remove $\tilde{\boldsymbol{k}}_q$ from the sum, continuing with the remaining features where we increase $t$ by one as the remaining features would capture higher-order interactions (and thus $|S|$ is increased by one). The second term, $\tilde{l}_i(q-1, t)$, generates the terms in the summation that exclude $\tilde{\boldsymbol{k}}_q$. In addition, the first condition, $\tilde{l}_i(i, t) = \tilde{l}_i(i-1, t)$, skips $\tilde{\boldsymbol{k}}_i$ since it is already accounted for in equation (11). The second condition, $\tilde{l}_i(0, t) = \frac{\sigma_t^2}{t}$, serves as the stopping criterion, ensuring that the weights $\sigma_t^2 / t$ of the Shapley value formulation are applied properly.

**Example 1** *We demonstrate the recursive formula for computing the Shapley value for the first feature for a case study of three feature. The recursive formula for the Shapley value is given by:*

$$\hat{\phi}_1^l = \left( \tilde{\boldsymbol{k}}_1 \bigodot \tilde{l}_1(3, 1) \right)^\top \boldsymbol{\alpha} \tag{12}$$

*Recursive steps are shown in Table 1. The last row of the table provides the Shapley value of the first feature, which is equivalent to equation* (9).

The recursive formula can be implemented by memorization in dynamic programming algorithms or alternatively by developing an iterative approach (Cormen et al., 2022). Both algorithms have the same time complexity of $O(d^2)$ for computing the Shapley value, which is a significant improvement over $O(2^d)$ of the crude computation of equation (9).

## 4 EXACT SHAPLEY VALUE FOR GLOBAL EXPLANATION

It is often important to measure the global importance of features in predicting output. Duvenaud et al. (2011) use interaction order variance to identify interactions at different orders, though their method faces the identifiability problem (Lu et al., 2022). A common approach to overcome this is global sensitivity analysis (GSA), which quantifies how much of the output variance is explained by each feature. Lu et al. (2022) introduced the first-order Sobol index and provided an analytical solution for its estimation. However, the first-order Sobol index only captures individual contributions and neglects interactions between features, limiting its ability to fully represent the importance of features in complex models (Song et al., 2016). Ironically, while AGPs are designed to model feature interactions, the Sobol index fails to account for them, which is a significant drawback. Shapley value, on the other hand, is shown to address the shortcomings adequately (Owen, 2014).

Table 1: Steps for computing the Shapley value of the first feature for an example of three features.

| Step | Calculation |
|---|---|
| 1. Compute $\tilde{l}_1(3,1)$ | $\tilde{l}_1(3,1) = \tilde{\boldsymbol{k}}_3 \odot \tilde{l}_1(2,2) + \tilde{l}_1(2,1)$ |
| 2. Compute $\tilde{l}_1(2,2)$ | $\tilde{l}_1(2,2) = \tilde{\boldsymbol{k}}_2 \odot \tilde{l}_1(1,3) + \tilde{l}_1(1,2)$ |
| | Use boundary condition: $\tilde{l}_1(1,3) = \frac{\sigma_3^2}{3}$ |
| | $\tilde{l}_1(2,2) = \tilde{\boldsymbol{k}}_2 \odot \frac{\sigma_3^2}{3} + \tilde{l}_1(1,2)$ |
| 3. Compute $\tilde{l}_1(1,2)$ | Use boundary condition: $\tilde{l}_1(1,2) = \frac{\sigma_2^2}{2}$ |
| | $\tilde{l}_1(2,2) = \tilde{\boldsymbol{k}}_2 \odot \frac{\sigma_3^2}{3} + \frac{\sigma_2^2}{2}$ |
| 4. Compute $\tilde{l}_1(2,1)$ | $\tilde{l}_1(2,1) = \tilde{\boldsymbol{k}}_2 \odot \tilde{l}_1(1,2) + \tilde{l}_1(1,1)$ |
| | Substitute: $\tilde{l}_1(1,2) = \frac{\sigma_2^2}{2}$ |
| | $\tilde{l}_1(2,1) = \tilde{\boldsymbol{k}}_2 \odot \frac{\sigma_2^2}{2} + \tilde{l}_1(1,1)$ |
| | Use boundary condition: $\tilde{l}_1(1,1) = \frac{\sigma_1^2}{1}$ |
| | $\tilde{l}_1(2,1) = \tilde{\boldsymbol{k}}_2 \odot \frac{\sigma_2^2}{2} + \frac{\sigma_1^2}{1}$ |
| 5. Substitute into $\tilde{l}_1(3,1)$ | $\tilde{l}_1(3,1) = \tilde{\boldsymbol{k}}_3 \odot \left( \tilde{\boldsymbol{k}}_2 \odot \frac{\sigma_3^2}{3} + \frac{\sigma_2^2}{2} \right) + \left( \tilde{\boldsymbol{k}}_2 \odot \frac{\sigma_2^2}{2} + \frac{\sigma_1^2}{1} \right)$ |
| 6. Compute $\hat{\phi}_1^l$ | $\hat{\phi}_1^l = \left( \frac{\sigma_3^2}{3} \tilde{\boldsymbol{k}}_1 \odot \tilde{\boldsymbol{k}}_2 \odot \tilde{\boldsymbol{k}}_3 + \frac{\sigma_2^2}{2} \tilde{\boldsymbol{k}}_1 \odot (\tilde{\boldsymbol{k}}_2 + \tilde{\boldsymbol{k}}_3) + \sigma_1^2 \tilde{\boldsymbol{k}}_1 \right)^\top \boldsymbol{\alpha}$ |

Owen (2014) showed the functional ANOVA decomposition holds true for the variance $\mathbb{V}$, i.e.,

$$\mathbb{V}(f(\boldsymbol{x})) = \sum_{S \subseteq D} \mathbb{V}(f_S(\boldsymbol{x})). \tag{13}$$

Accordingly, the following theorem states the relationship between the Shapley value for variance-based global sensitivity analysis.

**Theorem 1** *Let $\boldsymbol{\alpha}$ be the result of training an additive Gaussian process with a constrained kernel function. Then, the global Shapley value of feature $i$ based on the variance-based sensitivity analysis, shown by $\hat{\phi}_i^G$, is*

$$\hat{\phi}_i^G = \boldsymbol{\alpha}^\top \left( \sum_{S \subseteq D, i \in S} \frac{\sigma_{|S|}^4}{|S|} \Gamma_S \right) \boldsymbol{\alpha}, \tag{14}$$

*where $\Gamma_S = \odot_{i \in S} \int_{\mathcal{X}_i} \tilde{k}_i(\boldsymbol{x}, X) \tilde{k}_i(\boldsymbol{x}, X)^\top dp(x_i)$.*

The matrix $\Gamma_i, \forall i \in D$ can also be estimated with an empirical measure and proven to have an analytical solution with a Gaussian density of features (Lu et al., 2022). Also, the additive holds true for the global Shapley value as well with the value of the null game being zero, i.e.,

$$\mathbb{V}(f(\boldsymbol{x})) = \sum_{i=1}^{d} \hat{\phi}_i^G, \tag{15}$$

meaning that the Shapley value is the attribution of the variance of $f$ over the features.

Similarly to the local Shapley value, the challenge with computing the global Shapley value in equation (14) is that the summation includes an exponential number of terms. To address this, we first need to calculate $\Gamma_i$ for $i = 1, \ldots, d$, noting that $\Gamma_S = \odot_{i \in S} \Gamma_i$. Equation (14) can be written recursively, analogous to equation (9), as:

$$\hat{\phi}_i^G = \boldsymbol{\alpha}^\top \left( \Gamma_i \bigodot \hat{l}_i^G(d,1) \right) \boldsymbol{\alpha}, \tag{16}$$

where $\tilde{l}_i^G(q,t) = \Gamma_q \odot \tilde{l}_i^G(q-1, t+1) + \tilde{l}_i^G(q-1, t)$, with boundary conditions $\tilde{l}_i^G(i,t) = \tilde{l}_i^G(i-1,t)$ and $\tilde{l}_i^G(0,t) = \frac{\sigma_t^4}{t}$. This recursive formulation mirrors the interpretation of equation (11), with the key difference that $\hat{l}_i^G$ outputs a matrix, and the boundary conditions are adapted for the global Shapley value formulation in equation (14). However, the time complexity of the dynamic programming approach remains $O(d^2)$, offering a significant improvement over the brute-force complexity of $O(2^d)$.

## 5    RELATED WORK

Several local explainability methods leverage the Shapley value, with SHAP being a prominent example (Lundberg & Lee, 2017). SHAP provides additive explanations by attributing the predicted value to individual features. Model-specific extensions like Tree SHAP and Deep SHAP adapt this approach to decision trees and deep neural networks, and GP-SHAP offers local explanations for Gaussian Processes (Chau et al., 2023). Our method for local explanations in AGP is comparable to these techniques but is distinguished by its ability to compute exact Shapley values without requiring sampling. This eliminates the time-consuming and complex process of marginalizing over feature subsets, which is particularly challenging for tabular data and when features are interdependent.

In the context of global sensitivity analysis, our global Shapley value serves as a model-based alternative to the Shapley Effect (Owen, 2014; Song et al., 2016). The Shapley Effect uses Monte Carlo sampling to estimate Shapley values by generating a number of samples. In contrast, our method fits a model, specifically AGP, and computes exact global Shapley values without any reliance on sampling. This provides a more precise and efficient approach to global sensitivity analysis.

Our global Shapley value also aligns with methods like SAGE (Covert et al., 2020), which identifies the most important features within a trained model. It also serves as a tool for feature selection, akin to wrapper methods such as Lasso, which assess feature importance by first fitting a model. This capability is not only critical for model interpretability but also enhances the model's performance by focusing on the most relevant features.

Yet another similar line of research is functional decomposition-based interpretable models. The usefulness of functional decomposition for interpretability is discussed in the literature (Molnar, 2020, Chapter 8), and there are several studies showing the relationship between different interpretable models and functional decomposition (ANOVA) (Hiabu et al., 2023; Bordt & von Luxburg, 2023). However, the challenge is identifying exponentially many terms in functional decomposition. The AGPs, on the other hand, can handle the exponentially many terms as well as the constraints in the functional ANOVA decomposition in an efficient manner, and the proposed dynamic programming for local and global Shapley values provides an efficient way to compute Shapley values.

## 6    EXPERIMENTS

**Experimental Setup.**    In our experiments involving AGPs, we employ the squared exponential kernel function as the primary covariance function. To normalize the input data distribution, we incorporate normalizing flows (Rezende & Mohamed, 2015), and utilize the closed-form constrained squared exponential kernel as proposed by Lu et al. (2022). Due to the high computational complexity associated with training GPs, which scales as $O(n^3)$, we utilize inducing points for large-scale data sets to identify a subset of influential samples that effectively shape the AGP's decision boundaries (Burt et al., 2019). However, unless otherwise stated, all experiments are conducted using the complete set of available samples.

**Synthesized and Real Data Sets.**    We evaluate our approach using a combination of synthesized and real-world data sets. Detailed descriptions of the synthesized data sets and the generation process, as well as the characteristics of the real-world data sets, are provided in Appendix B. In experiments where inducing points are applied, Shapley values are computed exclusively from these points, with all other sample contributions set to zero in the $\boldsymbol{\alpha}$ vector. We use the same data sets in our experiments on feature selection and local explanation.

### 6.1    FEATURE SELECTION USING GLOBAL SHAPLEY VALUE

For feature selection, we evaluate the performance of the global Shapley value derived from the AGPs, referred to as AGP-SV, against several established feature selection methods, including HSIC-Lasso (Climente-González et al., 2019), mutual information (MI) (Vergara & Estévez, 2014), F-ANOVA (Koller et al., 1996), Lasso (Tibshirani, 1996), and recursive feature elimination (RFE) (Guyon et al., 2002).

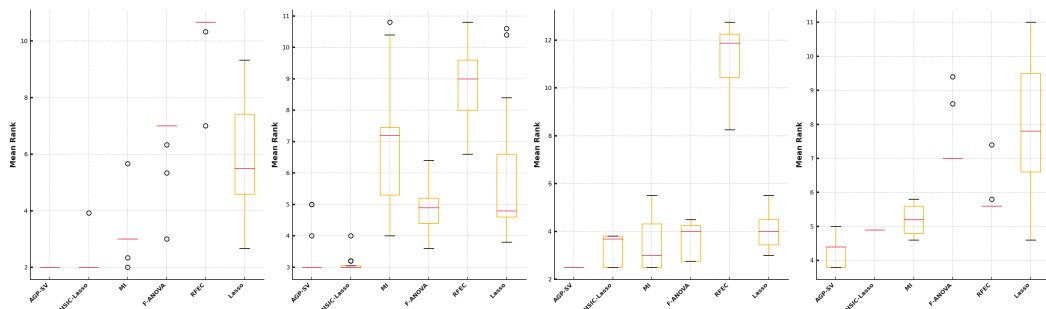

Figure 1: The comparison of feature selectors on synthesized four data sets; from left to right, the results over synthesized data set 1 to 4. The lower the mean rank, the better the results.

### 6.1.1 SYNTHESIZED DATA SETS

We generate four synthesized data sets with predefined influential features to benchmark the performance of different feature selection methods. Each data set is processed using the various feature selectors, and the identified influential features are compared across methods. To ensure an objective comparison, we compute the mean rank of the influential features based on the importance scores or rankings provided by each method. This feature selection process is repeated 100 times, and we calculate the average of these mean ranks to compare the methods comprehensively. Figure 1 presents box plots of the mean ranks across 100 replications for the synthesized data sets. AGP-SV consistently outperforms all other methods, demonstrating superior performance in identifying influential features. While HSIC-Lasso shows competitive performance on the first two data sets with simpler interaction functions (the ideal average of the mean rank is 2 and 3 for the first two data sets), owing to its ability to detect nonlinear relationships through kernel functions, AGP-SV significantly outperforms it when dealing with more complex interactions on synthesized data sets 3 and 4, reliably identifying the most influential features even in highly nonlinear settings (the ideal average of the mean rank is 2.5 and 3 for synthesized data sets 3 and 4).

### 6.1.2 REAL-WORLD DATA SETS

We extend our comparison of feature selectors to real-world data sets, utilizing tabular data sets such as "diabetes", "mode choice", "query", and "wine quality", with detailed descriptions provided in Appendix B. For the "wine quality" data set, we use 200 inducing points to fit the model and compute the Shapley value since the data set is large and expensive to train an AGP. To evaluate the effectiveness of the feature selectors, we first identify the most influential features using each method. Then, we incrementally train a random forest model with 500 estimators, sequentially adding features based on their importance rankings. We hypothesize that if the most influential features are added early in this process, the random forest's prediction error should decrease more sharply at the beginning and then level off as less important features are added. Figure 2 illustrates the error rate of the random forest as more features are included. In nearly all cases, AGP-SV consistently reduces the error more effectively than other feature selectors and demonstrates superior performance in identifying the most influential features across various real-world data sets. HSIC-Lasso demonstrates a competitive performance over the "diabetes" and "guerry" data sets, but AGP-SV shows more consistent performance across all data sets and outperforms HSIC-Lasso significantly.

## 6.2 LOCAL EXPLANATION USING LOCAL SHAPLEY VALUE

In this section, we evaluate the performance of the local Shapley value method against several state-of-the-art local explanation techniques. Specifically, we compare our approach to Kernel SHAP (Lundberg & Lee, 2017), Sampling SHAP (Štrumbelj & Kononenko, 2014), Unbiased SHAP (Covert & Lee, 2021), Bivariate SHAP (Masoomi et al., 2021), LIME (Ribeiro et al., 2016), and MAPLE (Plumb et al., 2018). All methods are used with their default settings, and we conduct comparisons on both synthesized and real-world data sets.

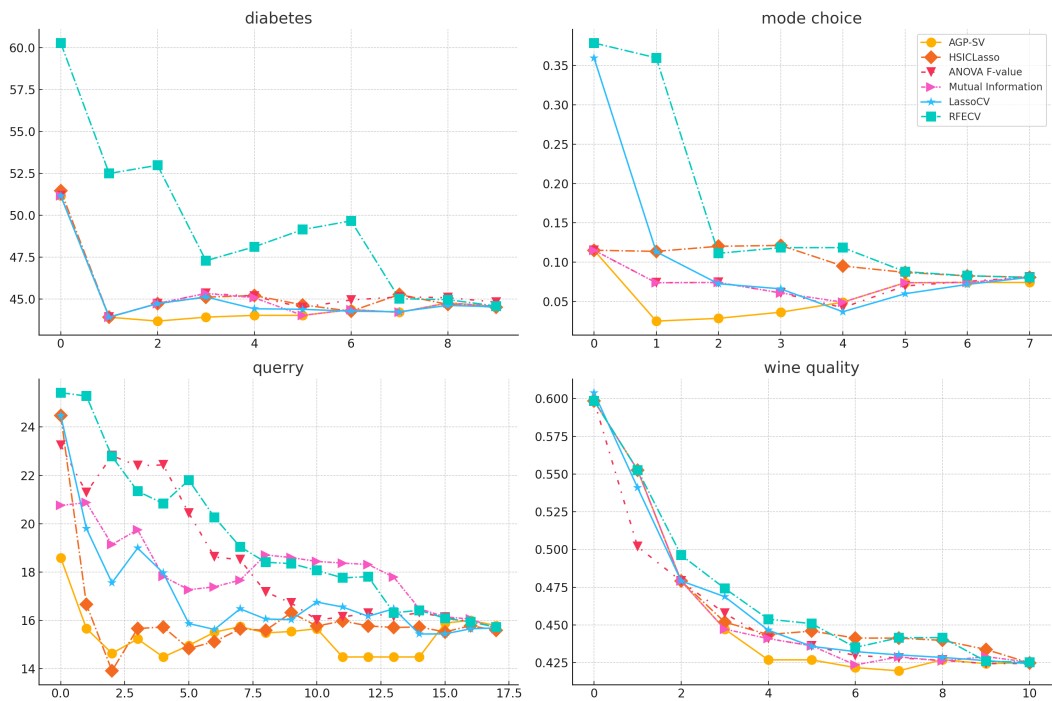

Figure 2: The performance of feature selectors over the real data sets.

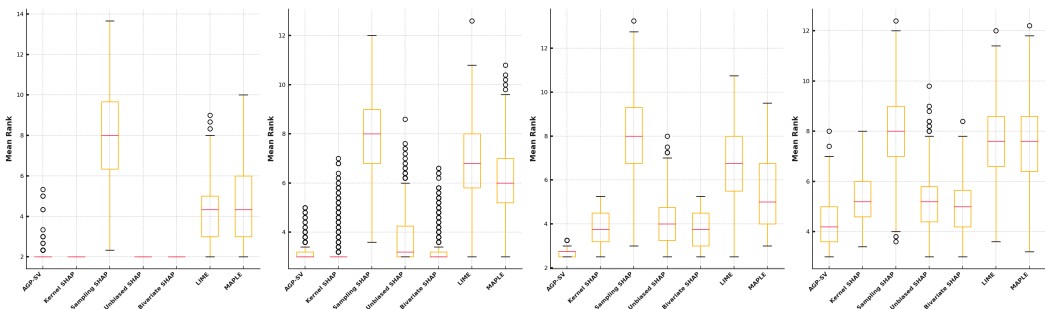

Figure 3: The average of mean rank of different explainable methods across four synthesized data sets (synthesized data sets 1 to 4 from left to right).

### 6.2.1 SYNTHESIZED DATA SETS

We begin by assessing the performance of the local explainers on four synthesized data sets (detailed information on the generation of these data sets can be found in Appendix B). For each data set, we randomly select 100 instances and apply each local explainer to generate explanations. We then calculate the mean rank of the most influential features identified by each method and compare the methods based on the average of these mean ranks across the 100 instances. Figure 3 presents box plots illustrating the performance of the local explainers across the four data sets. For the first two data sets (from left to right), AGP-SV demonstrates competitive performance with Kernel SHAP, Unbiased SHAP, Sampling SHAP, and Bivariate SHAP, particularly in cases where the interaction effects are more straightforward (average of mean rank is 2 and 3 for the first two data sets). However, in the last two data sets, where interactions are more complex, AGP-SV significantly outperforms the other methods, underscoring its robustness in capturing intricate interactions even at the local explanation level (the average mean rank is 2.5 and 3 for the last two data sets).

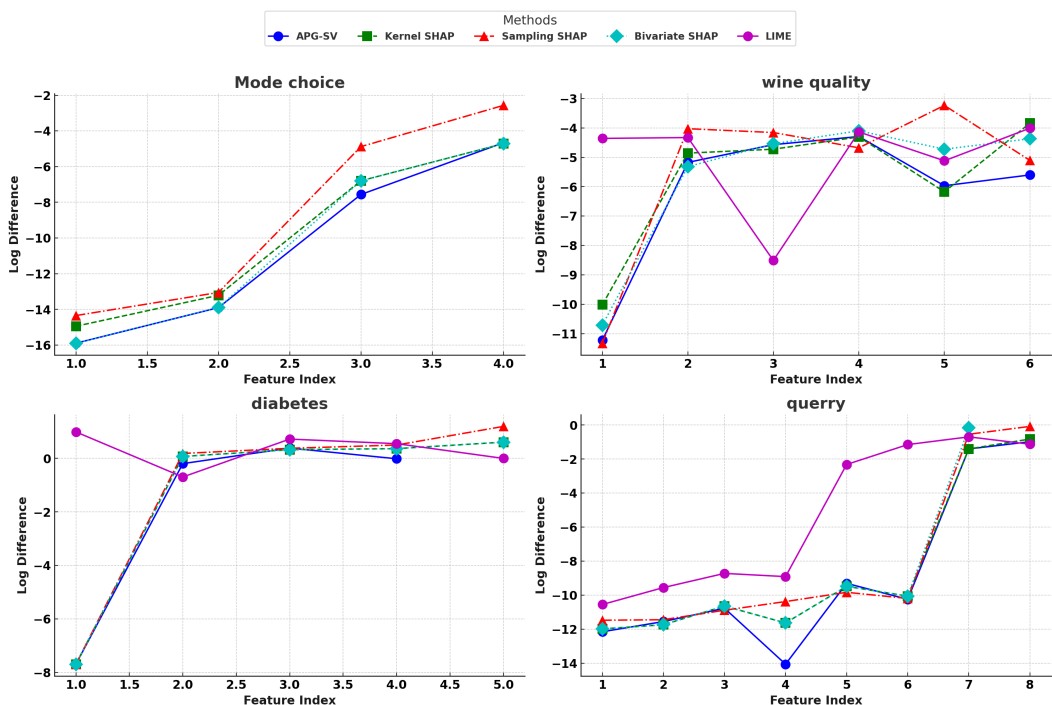

Figure 4: The effect of feature removal on the prediction of a trained AGP with constrained kernel over four real data sets.

### 6.2.2 REAL-WORLD DATA SETS

We extend our comparison of AGP-SV with other local explainers across four real-world tabular data sets. To conduct this comparison, we train an AGP with the constrained kernel and apply various explainable methods to it. The comparison is based on two metrics: the average execution time over 100 samples and the effect of incrementally removing the least influential features identified by each method. For the latter, we expect that the removal of the least influential features should have minimal impact on the AGP's predictions, particularly in the early stages of the process. Figure 4 shows the logarithm of the average impact of removing up to 60% of the features across 100 samples. We excluded Unbiased SHAP due to its inability to generate explanations for most samples, which made its average estimates unreliable, and MAPLE because its poor performance and large deviations from other methods made the plots difficult to interpret for small differences.

According to Figure 4, AGP-SV demonstrates consistent performance across all data sets, reliably identifying the most influential features for the 100 samples. For instance, in the "diabetes" data set, AGP-SV outperforms Kernel SHAP and Sampling SHAP in detecting the most influential features. Bivariate SHAP, which can detect interactions up to the second degree, also performs well and is competitive with AGP-SV in identifying the top influential features early on. However, AGP-SV shows a slight advantage in detecting the third most influential feature.

We also compared the methods based on the execution time required to generate explanations for 100 samples (for Unbiased SHAP, we only included the execution time for the samples where explanations could be generated). Figure 5 presents a box plot of the execution times for the different explainable methods. The results show that AGP-SV is significantly faster than the other methods while still providing exact Shapley values. In most cases, the exact Shapley value is calculated in around one second using the proposed dynamic programming. Coupled with its strong performance in identifying the most influential features, AGP-SV proves to be a highly efficient and effective method for feature importance detection.

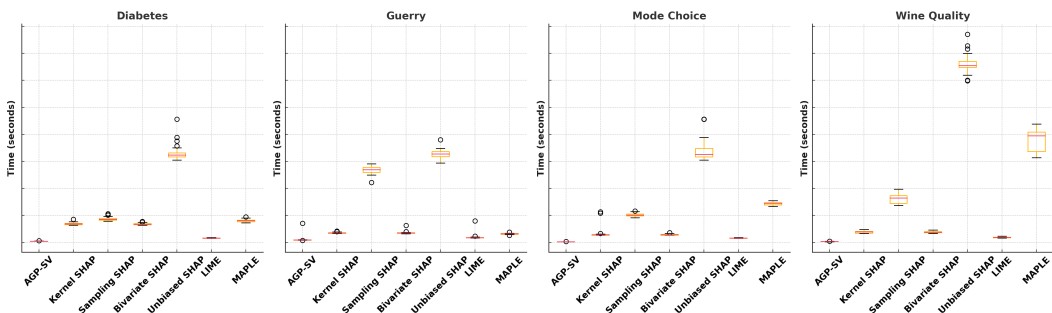

Figure 5: The box plot of the execution time in seconds for generating explanations for 100 samples.

# 7 CONCLUSION AND DISCUSSION

This paper presents a novel interpretative framework for Additive Gaussian Processes (AGPs) by leveraging the Shapley value to offer both local and global feature importance explanations. The main contributions include developing an efficient dynamic programming algorithm to compute the exact Shapley values, which scales polynomially with the number of features, and extending the variance-based global sensitivity analysis to AGPs for global explanations. Importantly, the AGP framework retains the predictive capabilities of standard GPs since the additive kernel structure subsumes typical GPs as a special case. The enhanced interpretability provided by the Shapley-based explanations makes AGPs a transparent and powerful tool in machine learning, especially for applications requiring both accurate predictions and clear, interpretable models.

There are some limitations and directions for future research. One key advantage of this approach is that it provides exact Shapley values, allowing for objective comparisons with other Shapley value-based explainable methods. This opens up the possibility of systematically varying parameters such as the number of samples to study how these methods behave under different conditions, thereby deepening our understanding of their robustness and reliability. Furthermore, while the current work focuses on tabular data, extending the AGP framework to other data modalities, such as text and image data, remains an important avenue for future exploration. Such extensions would broaden the applicability of AGPs and enhance their utility in diverse domains where interpretability is critical. Besides, while the focus of the paper is mainly on the Shapley value as the importance of individual features, further research could be done by identifying the most significant interactions among features and quantifying such quantities by relevant indicators such as the Shapley interaction index.

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

## A    PROOF

**Proof of Lemma 1**    To prove this lemma, we use some results from the functional ANOVA decomposition and the Shapley value. In particular, if we have the functional decomposition as in equation (1), then the Shapley value of feature $i$ for $\boldsymbol{x}$ is (Hiabu et al., 2023):

$$\hat{\phi}_i^l = \sum_{S \subseteq D} \frac{1}{|S|} f_S(\boldsymbol{x}) \tag{17}$$

Using the AGP, $f(\boldsymbol{x})$ can be written as:

$$f(\boldsymbol{x}) = K^{add}(\boldsymbol{x}, X)^\top \boldsymbol{\alpha}$$
$$= \left( \sigma_0^2 \mathbf{1} + \sum_{S \subseteq D, i \in S} \sigma_{|S|}^2 \bigodot_{j \in S} \tilde{k}_j(\boldsymbol{x}, X) \right)^\top \boldsymbol{\alpha}. \tag{18}$$

It follows that the AGPs provide the functional decomposition with the following elements:

$$f_0 = \sigma_0^2 \mathbf{1}^\top \boldsymbol{\alpha}$$
$$f_S(\boldsymbol{x}) = \left( \sum_{S \subseteq D} \sigma_{|S|}^2 \bigodot_{j \in S} \tilde{k}_j(\boldsymbol{x}, X) \right)^\top \boldsymbol{\alpha}, \tag{19}$$

then the Shapley value for feature $i$ is calculated as:

$$\hat{\phi}_i^l = \sum_{S \subseteq D, i \in S} \frac{1}{|S|} f_S(\boldsymbol{x}) = \left( \sum_{S \subseteq D, i \in S} \frac{\sigma_{|S|}^2}{|S|} \bigodot_{j \in S} \tilde{k}_j(\boldsymbol{x}, X) \right)^\top \boldsymbol{\alpha}, \tag{20}$$

and that completes the proof.

**Proof of Property 2**  To prove the additivity property, we can write:

$$\sum_{i=1}^{d} \hat{\phi}_i^l = \sum_{i=1}^{d} \left( \sum_{S \subseteq D, i \in S} \frac{\sigma_{|S|}^2}{|S|} \bigodot_{j \in S} \tilde{k}_j(\boldsymbol{x}, X) \right)^{\top} \boldsymbol{\alpha} = \left( \sum_{S \subseteq D, D \neq \emptyset} \sigma_{|S|}^2 \bigodot_{j \in S} \tilde{k}_j(\boldsymbol{x}, X) \right)^{\top} \boldsymbol{\alpha}. \quad (21)$$

By adding $f_0 = \sigma_0^2 \mathbf{1}^{\top} \boldsymbol{\alpha}$ to the above equation, we get the prediction function of the AGP, i.e.,

$$f(\boldsymbol{x}) = \sum_{i=1}^{d} \hat{\phi}_i + \sigma_0^2 \mathbf{1}^{\top} \boldsymbol{\alpha}. \quad (22)$$

$f_0$ can be interpreted as the value of the null game $v(\{\emptyset\})$ as well, and equation (22) indicates the additivity property of the explanation, and the proof is complete.

**Proof of Theorem 1**  Given that the functional ANOVA decomposition holds true for the variance $\mathbb{V}$, i.e.,

$$\mathbb{V}(f(\boldsymbol{x})) = \sum_{S \subseteq D} \mathbb{V}(f_S(\boldsymbol{x})), \quad (23)$$

then the global Shapley value based on the variance-based global sensitivity analysis is computed by:

$$\hat{\phi}_i^G = \sum_{S \subseteq D, i \in S} \frac{1}{|S|} \mathbb{V}(f_S(\boldsymbol{x})) \quad (24)$$

We now need to compute $\mathbb{V}(f_S(\boldsymbol{x}))$:

$$
\begin{aligned}
\mathbb{V}(f_S(\boldsymbol{x})) &= \mathbb{V}\left( \left( \sum_{S \subseteq D} \sigma_{|S|}^2 \bigodot_{j \in S} \tilde{k}_j(\boldsymbol{x}, X) \right)^{\top} \boldsymbol{\alpha} \right) \\
&= \sigma_{|S|}^4 \boldsymbol{\alpha}^{\top} \mathrm{cov}\left( \bigodot_{i \in S} \tilde{k}_i(x_i, X_i) \right) \boldsymbol{\alpha} \\
&= \sigma_{|S|}^4 \boldsymbol{\alpha}^{\top} \left( \bigodot_{i \in S} \int_{\mathcal{X}_i} \tilde{k}(\boldsymbol{x}, X) \tilde{k}_i(\boldsymbol{x}, X)^{\top} dp(x_i) \right) \boldsymbol{\alpha}.
\end{aligned}
\quad (25)
$$

Replacing equation (25) in equation (23), we get

$$\hat{\phi}_i^G = \boldsymbol{\alpha}^{\top} \left( \sum_{S \subseteq D, i \in S} \frac{\sigma_{|S|}^4}{|S|} \bigodot_{i \in S} \int_{\mathcal{X}_i} \tilde{k}(\boldsymbol{x}, X) \tilde{k}_i(\boldsymbol{x}, X)^{\top} dp(x_i) \right) \boldsymbol{\alpha}$$

and that completes the proof.

## B  DATA SETS

For our experiments on real data, we use the following tabular data sets:

- **Wine Quality:** This data set consists of 1,599 samples and 11 features related to the chemical properties of red wine. It is used to predict the wine quality score, which ranges from 0 to 10. The features include attributes such as acidity, sugar content, and alcohol levels.

- **Guerry:** The Guerry data set is a historical collection with 86 observations and 17 features, capturing various social statistics from 19th-century France. It includes data on literacy rates, crime rates, and donations to the poor, and is used primarily for exploratory data analysis and social science research.

- **Mode Choice:** The Mode Choice data set, derived from a transportation study, includes 840 samples and 6 features. It is used to model the decision-making process of individuals when choosing a mode of transportation, such as car, bus, or train, based on factors like income, travel time, and travel cost.
- **Diabetes:** This medical data set has 442 samples and 10 features, including age, sex, body mass index (BMI), and blood pressure. It is widely used for regression tasks to predict the progression of diabetes over one year.

In addition, we generate the following synthesized data in our experiments for explanation and feature selection:

- **Synthesized data Set 1**:
  Predictors $X$ are generated from a standard normal distribution. The response variable $Y$ is modeled as $Y = (X_1 X_2 X_3)$ (including a small noise factor), where only the first three features contribute significantly to the outcome. The expected mean rank for the most important features is ideally 2.

- **Synthesized data Set 2**:
  Predictors $X_1$ to $X_5$ are drawn from the standard normal distribution. The response variable $Y$ is defined such that $Y = 0.5 \left( \exp(X_1 X_2 X_3) + \exp(X_4 X_5) \right)$. Hence, only the first five features significantly influence the outcomes in this data set, with the expected mean rank ideally being 3.

- **Synthesized Data Set 3**:
  Predictors $X$ are similarly generated from the standard normal distribution. The response variable $Y$ is modeled so that $Y$ is proportional to $\exp \left( \sum_{i=1}^{4} X_i^2 - 4 \right)$, making the first four features particularly relevant. Therefore, the expected mean rank of the most important feature is ideally 2.5.

- **Synthesized Data Set 4**:
  Predictors $X$ follow a standard normal distribution. The response variable $Y$ is modeled as $Y = -10 \sin(0.2 X_1) + |X_2| + X_3 + \exp(-X_4 X_5)$, focusing on the first four features. As a result, the expected mean rank of the most important feature is ideally 3.

