# OpenReview forum: "Exact Shapley Value for Local and Global Explanation of Additive Gaussian Processes"
_ICLR.cc/2025/Conference — ICLR 2025 Conference Withdrawn Submission_

### Official Review · Reviewer_nVXm · 2024-10-24

**Soundness:** 3
**Presentation:** 3
**Contribution:** 3
**Rating:** 6
**Confidence:** 5

**Summary:**

High-dimensional modeling may be simplified by decomposing the initial function into a sum of components with increasing degree of interactions between the variables. For Gaussian process regression, additive models amounts to use additive covariance kernels. Identifiability is improved by adding some constraints, as proposed e.g., by (Lu et al., 2022). This paper shows that the corresponding model structure can be exploited to compute exact Shapley values efficiently. These may serve to identify important variables or groups of variables, for feature selection, either locally at a given x or globally by considering the variance. Shapley value estimation is tested on synthetic problems and data sets, against other Shapley value estimation methods.

**Strengths:**

- The proposed computation of Shapley value is exact, with reduced complexity compared to a direct approach.
- Global and local versions are proposed and tested on various examples

**Weaknesses:**

- The discussion of the Sobol indices for sensitivity analysis could be improved. Plus it is not added to the empirical comparison while available with the (Lu et al., 2022) approach.
- All quantities are computed on the predictive mean, what about exploiting the AGP predictive variance for better uncertainty quantification?
- The comparison of feature selection is solely based on the Shapley value.
- Some statements should be corrected, see questions below.

**Questions:**

On additive GPs and link with sensitivity analysis, see for instance:
- Durrande, N., Ginsbourger, D., & Roustant, O. (2012). Additive covariance kernels for high-dimensional Gaussian process modeling. In Annales de la Faculté des sciences de Toulouse: Mathématiques (Vol. 21, No. 3, pp. 481-499).
- Durrande, N., Ginsbourger, D., Roustant, O., & Carraro, L. (2013). ANOVA kernels and RKHS of zero mean functions for model-based sensitivity analysis. Journal of Multivariate Analysis, 115, 57-67.

P1, paragraph on first order Sobol index: what about total Sobol effect? Or derivative based sensitivity indices? Derivatives also provide information locally.

Beginning of Section 4: what about (Lu et al, 2022) where Sobol indices are computed for various orders of interaction on the posterior mean? Since they limit the number of terms, they can compute the total indices, by which they normalize. This baseline should be added to the empirical comparison.

P6: is the experimental setup using the code of (Lu et al, 2022), with inducing points and normalizing flows?

Section 6.1.2: what is the performance of a random forest with all predictors?

Figure 5: the time axis has no values

Conclusion: ``the additive kernel structure subsumes typical GPs as a special case’’ Why not showing the performance of a standard GP as a baseline then?

P7 : “guerry” or ‘‘query’’ data set? Or “querry” Figure 2 by cropping.

---

### Official Review · Reviewer_XThr · 2024-10-28

**Soundness:** 3
**Presentation:** 1
**Contribution:** 2
**Rating:** 1
**Confidence:** 3

**Summary:**

The paper introduces a quadratic programming algorithm that calculates the exact Shapley value as a decomposition of an additive Gaussian Process.

**Strengths:**

- Introduction was well written, the specific problem was discussed and the relevant related research was discussed to show the relevancy of the research question
- I liked how you differentiated your work from Chau et al. 2023 in the Related Work
- Good that you compare yourself to so many different works in the Experiments

**Weaknesses:**

### Background + Method
- The background assumes much knowledge of Gaussian Processes (GPs) to be able to follow it and doesn't introduce GPs at all
- I am missing any kind of intuition why Lemmas 1 and 2 hold (on the first read)
- The recursion "visualization" in Table 1 could be done via a figure in a much more understandable and compact way
- I dislike that Table 1 is so far away from Example 1
- I did not understand the relevancy of the global Shapley value

### Results
# **Let the y-axis begin at zero.**
I personally find this completely unacceptable except for very well explained exceptions as it gives me a sense of the authors trying to misrepresent their results to make them look better than they are.

- Figures 1 and 2 are borderline readable
- The captions of Figures 2 and 4 don't help me understand what the figures show. Figure 2 is also missing both x and y axis descriptions. Both are missing information whether smaller or larger values are better and how to read the results, generally.
- Figure 5 has no y-axis labels, but the x-labels are readable.
- I would prefer the figures to be vector graphics, I recommend the authors to consider LaTeX tikz to create visualizations.
- An experiment I am missing due to the statements in your paper would be a, perhaps small scale, comparison to the naive Shapley value calculation. While you mention several times that you're _exact_, it would make a strong point highlighting this in one to two cases by showing that the error is perhaps $1e-6$ or less.

### Appendix
- Synthesized data Set 2 has either a wrong or at least unfortunately formulated description in Appendix B. And why do you say "the first five features" when there's only $X_1$ through $X_5$?
- I generally fail to understand the mean ranks that you provide for the synthesized data sets and can't find an explanation close to the statements.

**Questions:**

- You mention several times that you precisely receive the Shapley value using your approach, yet Lemma 1 speaks about the "estimated" Shapley value?
- Does your approach still hold for mean functions $\mu(x) \neq 0$?
- Typo in Equation (4)? You have $\int k_i(x, s) dp(s)$ twice in the numerator.

---

### Official Review · Reviewer_VfWP · 2024-10-29

**Soundness:** 3
**Presentation:** 2
**Contribution:** 2
**Rating:** 3
**Confidence:** 3

**Summary:**

In this submission, the authors enhance a class of interpretable models, Additive Gaussian Processes (AGPs), with a well-known \emph{post hoc} explainability tool inspired by game theory: Shapley values (SVs). AGPs by themselves are not completely interpretable due to known identifiability issues, which have been accounted for by Orthogonal Additive GPs (OAK) [1]. However, even if there exist methods for computing SVs for GPs [2], these do not leverage the precise structure of OAK-GPs. Here, the authors remedy this issue and demonstrate that SVs can be formulated for OAK-GPs, and that these additive SVs inherit the properties of usual SVs.
One of the paper's main contributions is deriving a dynamic programming scheme for the computation of \emph{exact} Shapley values, both in the local and global setting and at a cost that scales quadratically with the dimensions, instead of exponentially. Standard methods which do not leverage the particular structure of the model like KernelShap only compute approximated Shapley values and take substantially longer to do so.
Finally, experiments on both synthetic and real-world datasets illustrate the soundness of the recovered additive Shapley values.

Although the paper tackles a niche problem, explaining Additive Gaussian Processes, the execution is theoretically grounded, efficient, and seems to produce relevant explanations on synthetic examples. However, I have several concerns regarding the experiments section, that I express in the weaknesses section. Currently, I would not recommend the paper for acceptance, not before clarification of these concerns.

[1] Orthogonal additive Gaussian processes - ICML 2022
[2] Explaining the uncertain: Stochastic Shapley values for Gaussian process models - NeurIPS 2023

**Strengths:**

- The method is theoretically grounded and an effort has been made to provide fast computation algorithms, reducing the naive exponential complexity to a quadratic complexity.

- I liked that the authors included an example of how to compute AGP-SV for 3 features, illustrating the dynamic programming algorithm (Table 1).

**Weaknesses:**

Although the theoretical and algorithmic aspects of the paper make it a promising contribution, I found the experiment to be misleading. I acknowledge that this might be due to a poor understanding on my side, but the concerns I will now raise are enough for me to recommend rejection of the paper.

- My understanding is that the proposed method gives a principled way to compute shapley values for additive GPs, whereas methods like KernelShap or GPshap will require approximations, and do not leverage the additive structure of the model.
As such, the main way to evaluate the method should be on synthetic examples for which an ordering of the global/local feature relevance is known, against different SHAP-based alternatives. The most accurate method would recover the ordering that is the closest to the true one.
Section 6.1.1 does such analysis, but involves feature selection methods like Mutual Information, HSIC-Lasso, Lasso, etc. I do not think this section provides any insights, rather, it can be misleading, as the feature selection process depends on a model that is different for each method. For instance, Lasso employs a linear model, HSIC-Lasso depends on a specific kernel, etc.
At the very least, the setting employed for each method should be described in the appendix, e.g. what is the kernel for HSIC-lasso, what is the sparsity hyperparameter value for Lasso?

- Next, AGP-SV is compared against the same feature selection methods on real-world datasets. As no ground truth is available, selected features are used to build a random forest model, with the assumption that the right ordering of feature relevance will lead to the best decrease in prediction error as relevant features are imputed. I do not think that this experiment provides any insights about AGP-SV. If anything, it only demonstrates that Additive Gaussian Processes themselves are a suitable model and that it leads to extracting more relevant features than Lasso would, for instance. It is my understanding that disentangling the suitability of the additive GP model itself, with how Shapley values are computed for that particular model, is not straightforward.
Moreover, the appendix should provide details on how is the prediction error computed here. Using the Out-of-bag estimate of the RF model? On a holdout test set?

- In the comparison with other SHAP-based methods (section 6.2), why not include GPShap, a method tailored for GPs (but not additive GPs), mentioned in the related work?

- An evaluation against high-dimensional datasets would have been beneficial. Either using real-world datasets or synthetic ones. Currently, the largest example is of dimensionality $d=17$. A synthetic example with, say, $d=50$ and an actual number of relevant features of $p=5 \ll d$ would have been interesting. Actually, the synthetic dataset section mentions "influential features", like $X_1, X_2 ,X_3$ for the first example, but it is never mentioned how many features are there \emph{in total}. Could the authors clarify this point?

**Questions:**

- As the probabilistic counterpart of Kernel methods, Gaussian Processes provide principled uncertainty quantification, which is accessible in closed form for Additive GPs. I understood that the proposed aims at explaining the mean prediction of the GP $f(\mathbf{x})$, but it is not completely straightforward if the available prediction uncertainty is even being used. Could the authors clarify this point, although I just might have missed something?

- Also see the weaknesses section.

A few typos appear throughout the manuscript:

- Eq 4: the second integral in the numerator should involve $\mathbf{x}'$, not $\mathbf{x}$.
- Figure 2: "guerry" instead of "query".
- Appendix, synthesized data set 4: "focusing on the first _five_ features"

---

### Official Review · Reviewer_yuJQ · 2024-10-30

**Soundness:** 2
**Presentation:** 4
**Contribution:** 3
**Rating:** 3
**Confidence:** 4

**Summary:**

The authors derive an exact Shapley value for both local and global explanation of individual feature importance in additive Gaussian processes with Gaussian likelihood. While naive computation of the Shapley value would be $\mathcal{O}(2^d)$ for $d$ features, they present a recursive dynamic programming algorithm that instead runs in only $\mathcal{O}(d^2)$. The proposed feature importance metric is evaluated on four synthetic data sets and four small real-world data sets. The authors compare performance both against other feature importance methods (LIME, MAPLE, various SHAP-based methods) and performance as a feature selection metric (in comparison to baselines such as Lasso). In the author’s execution time comparison, the proposed method is the fastest amongst all local feature importance methods included.

**Strengths:**

Originality: The present work is clearly original; the authors derive an exact formulation of Shapley value in the additive Gaussian process model and present a clever recursive formulation that allows for an efficient implementation using dynamic programming.

On the theoretical side, the writing is clear and fairly easy to follow. I appreciated the explicit worked-through example for the dynamic programming.

The resulting method seems to be very fast empirically, and seems to be a significant contribution to the evaluation/analysis of additive Gaussian process models.

**Weaknesses:**

The paper’s key motivation is that for the AGP model there is no native method that provides local explanations and that the global explanations have notable shortcomings.

Regarding the shortcomings of existing global explanations, the paper claims that the Sobol index (with an analytical computation provided in Lu et al. (2022)) “neglects interactions between features”; however, Lu et al. compute Sobol indices for every feature _set_, not just individual features (as shown in Lu et al.’s Figure 5, they do add up to one). Whether it is preferable to incorporate the interaction terms in a feature’s individual importance might be a base for fruitful discussion, but does not seem immediately obvious to me.

Unfortunately, the paper’s empirical evaluation is very weak. I would have liked to see an evaluation on a similar level to Lu et al. (2022), who evaluated their proposed method on 24 regression and 29 classification data sets (from a hundred to 45 thousand data points and from 3 to over a hundred dimensions), and provided several detailed case studies into the model as well as the Sobol indices for feature importance. This paper only considers 4 real-world data sets with at most 1600 data points and 17 dimensions. (Without clear arguments for why these specific data sets, such a small-scale evaluation makes the reader wonder whether the data sets were cherry-picked - even if the four data sets were selected completely arbitrarily! Better to use a comprehensive range of standard data sets, and/or explain how you arrived at your choice.) Moreover, it only compares the ‘downstream’ performance as a feature selection method.

It is not clear from the experiments how the local explanations differ from the global explanation. It would be better to be able to show how different instances have different local explanations/feature importances, and ideally compare some analytic solution against the baselines and proposed method. This is also an area where a careful case study (real-world data set, actually look at individual instances, their predictions, features important for this prediction, visualize the instance on the model's feature and interaction components - as that's what AGP is made for) would significantly strengthen the paper.

The paper mentions “objective comparisons with other Shapley value-based explainable methods” in the discussion section, but there is no evaluation of Shapley values themselves, and no direct comparison to the Sobol values that can also be computed analytically.

In the related work section, GP-SHAP is mentioned as an approach specialized to Gaussian processes; this would seem to be a very relevant baseline, but it is not included in the experiments.

The paper does not provide sufficient details to be replicable. For example, for synthetic data sets, there is no indication for how many data points were generated, how many “non-influential” dimensions there are beyond the three to five relevant features (to what extent would this affect the results? this would be an important, yet missing experiment), what the distribution of the noise factor is. For real-world data sets, there is no indication where the data sets are from. There is no indication if code will be made available.

Overall, I think this paper has a high potential, it just does not seem to be ready yet. I would strongly encourage the authors to improve the empirical evaluation; with that, I would expect the paper to have a high chance at a future submission to e.g. ICML or NeurIPS 2025. (Those might also be better-fitting venues for this work, ultimately giving it wider reach. (In general it's more of a topic for something like AISTATS than deep learning-focused ICLR, but I would have given the same review there.))

One non-empirical point: as far as I can tell, the proposed approach only directly applies to (additive) Gaussian process _regression_, and would not extend to classification or other non-Gaussian likelihoods; this limitation is not clear from the manuscript. (Note, this is about being explicit about assumptions/limitations/requirements, not about "your method can't do X".)

**Questions:**

1. “AGPs struggle to provide local explanations and offer only global feature importance”: if I understand correctly, this is not something about AGPs as a model - in fact, your paper introduces local explanations for AGPs - so it would seem more appropriate to write something along the lines of “So far, there had not been any method for deriving local explanations from an AGP model” – though, could one have used for example some gradient-based attribution method out of the box?

2. “[AGP’s] additive structure enhances the model’s ability to capture complex, higher-order interactions” – this does not seem right as is; a GP with a squared-exponential kernel can capture complex, higher-order interactions just the same. The key advantages of the additive structure (including the higher-order kernels for capturing interaction terms) is the interpretability.

3. The notation is sometimes non-standard; I would suggest to use standard notation such as `\mathbb{R}` instead of `R` for the space of real numbers.

4. What do you mean by “sample space of feature subset $S$”?

5. The paper uses some of the notation from the origins of Shapley value in game theory, such as “coalition $S$”, “null game $v(\{\emptyset\})$”, “realizing $v$”; for a paper aimed at machine learning audience, it would be helpful to omit these if not needed, or to connect better the machine learning term (e.g. “feature subset $S$, corresponding to ‘coalition’ in theory of Shapley values”).

6. Can you give some intuition why the choice in Eq. (5) is appropriate?

7. p.5 above Eq. (15): “the additive holds true” – what does this mean?

8. Example 1/Table 1: While $\tilde{l}_1(1, t)$ is one boundary condition ($i=q$), it would be helpful to explicitly include the intermediate step of $\tilde{l}_1(0, t)$ (boundary condition $q=0$). Unfortunately, $i=1$ is an edge case in itself; it might more clearly show the different steps to choose $i=2$ instead for the example walk-through.

9. You mention GP-SHAP: why do you not compare against this? It might be instructive to compare the produced Shapley values directly between methods. (Similarly for Shapley Effect.)

10. Computational details: how did you select the inducing points? How does the number and choice of inducing points affect the computed Shapley value ?

11. Figs. 1 and 3: the labels are extremely difficult to read, would be better to use a larger font size.
Fig. 2: would be easier to follow with some labels and more explanation in the caption.

12. Figure 2: what is the x-axis? What happens at x=0, and why would the different methods already perform differently?

13. “in the last two data sets, where interactions are more complex, AGP-SV significantly outperforms the other methods”: I would agree to ‘significantly’ only for the 3rd dataset; for the fourth dataset, three of the SHAP methods perform similarly.

14. “Figure 4 shows the logarithm of the average impact”: how exactly is this computed? In Fig. 4 it is referred to as “log difference”; it would be helpful if it was consistent with the text. For this metric, presumably ‘lower is better’, but again be helpful to explicitly mention this in the figure.

15. “Unbiased SHAP [is unable] to generate explanations for most samples”: what does ‘inability’ mean: the code crashes? Could you clarify?

16. “MAPLE[‘s] poor performance and large deviations made the plots difficult to interpret”: still preferable to have the full plots for all methods in the appendix.

17. Execution time, Fig. 5: How does it depend on the model/dataset? Am I correct to assume it would be linear in the number of data points for all methods, and the key difference is in prefactors and dependence on number of features? If so, then a more useful presentation would be to consider datasets of varying $d$ (can also be on synthetic datasets – presumably it would be the same for same combination of $n$/$d$), and then plot number of features (dimensionality) on x axis and time on y axis and the different methods as separate lines.

18. Assuming the “mode choice” dataset refers to https://www.statsmodels.org/stable/datasets/generated/modechoice.html, this seems to be categorical/classification labels, how did you handle this with the regression model?

---

### Note · Authors · 2024-11-26

I have read and agree with the venue's withdrawal policy on behalf of myself and my co-authors.